# Association of Regional Bone Synthetic Activities of Vertebral Corners and Vertebral Bodies Quantified Using ^18^F-Fluoride Positron Emission Tomography with Bone Mineral Density on Dual Energy X-ray Absorptiometry in Patients with Ankylosing Spondylitis

**DOI:** 10.3390/jcm9082656

**Published:** 2020-08-17

**Authors:** Keunyoung Kim, Kyoungjune Pak, In-Joo Kim, Seong-Jang Kim, Dong Hyun Sohn, Aran Kim, Seung-Geun Lee

**Affiliations:** 1Department of Nuclear Medicine, Pusan National University Hospital, Busan 49241, Korea; nmpnuh@gmail.com (K.K.); ilikechopin@pusan.ac.kr (K.P.); injkim@pusan.ac.kr (I.-J.K.); 2Biomedical Research Institute, Pusan National University Hospital, Busan 49241, Korea; solees17@naver.com; 3Department of Nuclear Medicine and Research Institute for Convergence of Biomedical Science and Technology, Pusan National University Yangsan Hospital, Yangsan 50612, Korea; growthkim@pusan.ac.kr; 4Department of Microbiology and Immunology, Pusan National University School of Medicine, Yangsan 50612, Korea; dhsohn@pusan.ac.kr; 5Divsion of Rheumatology, Department of Internal Medicine, Pusan National University School of Medicine, Pusan National University Hospital, Busan 49241, Korea

**Keywords:** ankylosing spondylitis, positron emission tomography, bone density, cancellous bone, cortical bone

## Abstract

We investigated whether the bone-synthetic activities of vertebral bodies or vertebral corners quantified using ^18^F-fluoride positron emission tomography (PET) was associated with bone mineral density (BMD) at the corresponding lumbar vertebrae in ankylosing spondylitis (AS) at each vertebra level. We analyzed 48 lumbar vertebrae in 12 AS patients who underwent ^18^F-fluoride PET and dual energy X-ray absorptiometry (DXA). The mean standardized uptake values (SUVmean) of the vertebral body and corners from L1 to L4 were measured using the spatially separated region of interest (ROI). The L1–L4 BMDs were calculated based on the DXA (“conventional BMD”). The BMD of the internal vertebral bodies was measured by manually drawing ROIs to represent the trabecular BMD (“alternative BMD”). After adjusting the within-patient correlation, the ^18^F-fluoride SUVmean of the vertebral corners but not that of vertebral bodies was significantly related with the conventional BMD of the vertebra. Otherwise, the ^18^F-fluoride uptake of both the vertebral and vertebral bodies was significantly related with the alternative BMD. The bone-synthetic activities of the vertebral corners may be more closely related with BMD than those of the vertebral bodies, suggesting that the effects of regional bone metabolism at the vertebral corners and bodies on BMD differ in AS.

## 1. Introduction

Ankylosing spondylitis (AS) is a chronic rheumatic disease characterized by inflammation in the axial skeleton such as the sacroiliac joints and spine, subsequently leading to new bone formation. In AS, excessive osteoproliferation such as syndesmophytes and complete ankylosis (bony bridging) at anterior vertebral corners arises from the local cortical bone compartment and results in the limitation of spinal mobility and a loss of function [1]. Trabecular bone loss leading to osteoporosis of the spine and hip is another prominent characteristic of AS [2] and can occur even in the early stage of the disease [3], which predisposes patients with AS to a higher risk of osteoporosis and fragility fractures compared to that for the general population [4,5,6,7]. Thus, in AS, there is an apparent paradox of excessive bone formation and bone loss at anatomical sites closely located to each other; moreover, the regional bone metabolism between the vertebral corners (cortical bone) and bodies (trabecular bone) may differ in relation to the bone mineral density (BMD) of the vertebrae. The underlying mechanism therefore remains elusive. To identify this issue, quantitative assessment of bone metabolism in specific sites of the vertebrae may be necessary.

^18^F-fluoride-labelled positron emission tomography (PET) allows the noninvasive quantitative assessment of the regional bone metabolism at different sites of the skeleton by measuring standardized uptake values (SUVs) [8,9,10]. The advantages of ^18^F-fluoride PET over conventional 99mTc-labelled phosphate and diphosphonate bone scintigraphy include higher image quality, sensitivity, specificity, and diagnostic accuracy as well as a shorter examination time [11] along with the ability to distinguish metabolic changes between cortical and trabecular bones [8,12,13]. As animal and human studies have provided evidence that ^18^F-fluoride uptake indicates active osteoblastic bone synthesis [14,15], the clinical usefulness of ^18^F-fluoride PET has been recently documented for evaluating site-specific bone formation in various skeletal disorders including osteoporosis and AS [8,16,17]. Regional bone metabolism assessed using ^18^F-fluoride PET was found to correlate strongly with BMD and biochemical bone turnover markers in patients with osteoporosis [18]. We recently reported that an increase in ^18^F-fluoride uptake at the anterior vertebral corners on a PET scan was associated with current and future syndesmophyte development in patients with AS [19,20]. Taken together, ^18^F-fluoride PET may be a suitable tool for quantitatively and separately measuring the regional bone metabolism at the vertebral corners and bodies and may have a potential role in investigating the complex pathophysiology of bone metabolism in AS.

Therefore, in the present study, we assessed the magnitude of bone formation at the vertebral corners and bodies, respectively, using ^18^F-fluoride PET in patients with AS and examined whether the baseline ^18^F-fluoride uptake of the vertebral corners or bodies on PET is associated with BMD at the corresponding lumbar vertebrae at each vertebral level in this population.

## 2. Experimental Section

### 2.1. Study Design and Subjects

This study was a post hoc analysis of prospectively gathered data obtained from previous reports [19]. In our previous study, 12 male patients with AS who fulfilled the modified New York Criteria [21] were consecutively recruited and underwent ^18^F-fluoride PET-magnetic resonance imaging (MRI) at baseline in the previous study [19]. Dual energy X-ray absorptiometry (DXA) in these 12 AS patients was conducted at baseline for the present study. Because DXA can only measure the BMD of L1 to L4 vertebrae, we only used PET data of the L1 to L4 vertebrae in the present study, whereas PET data of the whole spine were used in the previous study. BMD data obtained from DXA examination were only used for the present study. All the study subjects were naïve to biologic agents such as tumor necrosis factor-alpha (TNF-α) inhibitors. The following subjects were excluded from the study: (1) patients with musculoskeletal disorders other than AS; (2) subjects with current or previous use of medication affecting bone metabolism including calcium, vitamin D, bisphosphonates and teriparatide; (3) previous spine surgery or metal implant patients; (4) patients who refused to participate in the study. The Research and Ethical Review Board of the Pusan National University Hospital approved this study (H-1401-017-014), and all patients provided their written informed consent based on the Helsinki Declaration.

Clinical data in patients with AS including age, disease duration, body mass index (BMI), serum erythrocyte sedimentation rate (ESR), C-reactive protein (CRP), the Bath Ankylosing Spondylitis Disease Activity Index (BASDAI), the Bath Ankylosing Spondylitis Functional Index (BASFI), the Bath Ankylosing Spondylitis Metrology Index (BASMI) and concomitant medications were collected. Serum CRP levels were measured by the particle-enhanced immunoturbidimetric assay (Tinaquant C-reactive protein; Roche Diagnostics, Rotkreuz, Switzerland) with an automated analyzer (P-800 Modular; Roche Diagnostics). BMI was determined as weight in kilograms divided by the square of height in meters. The anteroposterior and lateral views of the cervical and lumbar spine radiographs were obtained. Lateral radiographs of the cervical and lumbar spine were scored using the modified Stoke Ankylosing Spondylitis Spine Score (mSASSS) by two rheumatologists blinded to the patient details. The anterior corners of C2 lower to T1 upper and T12 lower to S1 upper were scored for the presence of squaring, sclerosis and/or erosion (1 point), a non-bridging syndesmophyte (2 points) and a bridging syndesmophyte (3 points) [22]. Only the mSASSS data that were concordant between the two readers were used in our data as recommended previously [23].

### 2.2. Imaging Protocol

The present study used the ^18^F-fluoride PET images obtained at baseline from the ^18^F-fluoride PET-MRI image acquired from the Philips Ingenuity TF sequential whole-body PET-MRI system (Philips Healthcare, Cleveland, OH, USA). PET scans from the top of the head to the thighs were obtained 60 min after the intravenous injection of a dose of 245–285 MBq of ^18^F-fluoride. The effective dose of ^18^F-fluoride PET for the patients ranged from 6.5 to 7.5 mSv. The emission scan time per bed position was 3 min; 9 bed positions were acquired. PET data were obtained using a high resolution whole body scanner with an axial field of view of 18 cm. The average axial resolution varied between 4.2 mm full width at half maximum in the center and 5.6 mm at 10 cm. The average total PET-MRI examination time was 30 min. After scatter and decay correction, the PET data were reconstructed iteratively with attenuation correction and reoriented in axial, sagittal and coronal slices using the pre-performed CT for the clinical evaluation of the patients within 1 week of the PET. The row action maximum likelihood algorithm was used for 3-dimensional reconstruction. The reconstructed images were converted to SUV images based on following equation: SUV = tissue activity (kBq/mL) × body weight (kg)/injected ^18^F-fluoride dose (MBq).

At baseline, DXA was conducted to quantify BMD using a single DXA scanner (GE-Lunar Prodigy, GE, Madison, MA, USA). The entire lumbar spine was scanned in the supine position using posteroanterior projections, and the BMD at the lumbar spine was calculated for the first to fourth vertebrae using densitometric software (Encore software, version 13.0, WI, USA). Daily calibration and quality assurance tests were performed, and the coefficient of variation for the precision of the repeated DXA measurements was 0.32%.

### 2.3. Imaging Analyses and Interpretation 

A nuclear physician who was blinded to the clinical data analyzed the PET images. ^18^F-fluoride uptake was measured using the Philips Extended Brilliance Workspace version 4.5.3.40140 (Philips Healthcare, Best, The Netherlands) using the mean SUV (SUVmean). Using the rectangular region of interest (ROI), the SUVmean of the vertebral bodies was acquired according to a previous study performed by Uchida et al. (Figure 1A) [24]. For the SUVmean of the vertebral corners, we drew a circle of 10 mm in diameter, and we did the same for the SUVmean of the upper and lower vertebral corners and calculated the arithmetic mean SUVmean of those that were used for the analysis (Figure 1B). All ROIs for the SUVmean of the vertebral body and both upper and lower vertebral corners from the L1 to L4 were drawn in the mid-sagittal plane of the ^18^F-fluoride PET images. We decided that the sagittal plane was more suitable for the analysis with the projection image of the DXA than the trans-axial plane of the PET. Additionally, both vertebral corners and vertebral body activities could be acquired simultaneously in the mid-sagittal plane, which seems more apt for the comparison of two different sites of vertebra at the same time. We also measured and analyzed the maximal SUV (SUVmax) of lumbar vertebrae. The SUVmean and SUVmax were measured five times per patient, and the intra-operator correlation coefficients were 0.9964 (95% confidence interval, 0.9945–0.9978) and 0.9991 (95% confidence interval, 0.9987–0.9995) for SUVmean and SUVmax, respectively.

We used two different ROIs for calculate the BMDs of L1–L4 from the DXA images, expressed in g/cm^2^. The ROI-1 was defined as the whole BMD of each vertebra, which is used in the routine conventional clinical setting (Figure 1C). The other ROI-2 was devised for excluding the cortical bone area, which was assumed to represent the trabecular bone area, separately (Figure 1D). BMD measured using ROI-1 and ROI-2 was termed as “conventional BMD” and “alterative BMD”, respectively.

### 2.4. Statistical Analyses

The results are presented as the mean ± standard deviation (SD) or median (interquartile range (IQR)) for normally distributed or non-normally distributed continuous variables and as number (percentage) for categorical variables, as appropriate. The Kolmogorov–Smirnov test was used to determine the normality of the data. For group comparisons, Student’s t-test was used for continuous variables and Pearson correlation analyses were conducted to estimate the correlations between the ^18^F-fluoride SUVmean and BMD at each vertebra level. Because the lumbar vertebra level data were correlated (or clustered) within each patient, multilevel mixed-effects linear regression models adjusting for within-patient correlation for the total number of lumbar vertebrae were performed using the xtmixed command of STATA 11.1 for Windows in order to assess the association among ^18^F-fluoride SUVmean and BMD. In particular, to analyze the independent association between conventional BMD and alternative BMD (outcome variables) and the ^18^F-fluoride SUVmean of the vertebral corners and bodies (independent variables), multivariate multilevel mixed-effects linear regression models including variables with *p* < 0.2 in univariable models were used. We also analyzed the association between ^18^F-fluoride SUVmax and BMD in the same way as SUVmean. All statistical analyses were performed using STATA 11.1 for Windows (StataCorp LP, College Station, TX, USA), and *p* ≤ 0.05 was considered statistically significant.

## 3. Results

The demographic and clinical data of the 12 male patients with AS are presented in Table 1. The median (IQR) age and disease duration were 42 (32–42.8) and 7.5 (2.4–10.1) years, respectively. The median (IQR) mSASSS and BASDAI were 9 (5.3–28) and 3.5 (1.2–6), respectively. All patients had a positive result for HLA-B27.

A total of 48 lumbar vertebrae from 12 patients with AS were analyzed in this study. Lumbar vertebral level data such as the ^18^F-fluoride SUVmean of the vertebral corners and bodies on PET and the BMD of the vertebra and vertebral body on DXA were normally distributed according to the Kolmogorov–Smirnov test and are summarized in Table 2. There was no difference in the ^18^F-fluoride SUVmean between the vertebral corner and vertebral body (5.24 ± 0.9 vs. 5.03 ± 0.9, *p* = 0.239) whereas the conventional BMD of the vertebra was significantly higher than the alternative BMD of the vertebra (1.19 ± 0.21 vs. 1.03 ± 0.22, *p* = 0.001).

The results of the simple correlation analyses are depicted in Figure 2. The ^18^F-fluoride SUVmean of the vertebral corners was positively correlated with that of the vertebral bodies (γ = 0.716, *p* < 0.001). The conventional BMD of the vertebra was positively correlated with the ^18^F-fluoride SUVmean of the vertebral corners (γ = 0.402, *p* = 0.005) but not with that of the vertebral bodies (γ = 0.279, *p* = 0.055). Otherwise, the alternative BMD of the vertebra was positively correlated with the ^18^F-fluoride SUVmean of both vertebral corners (γ = 0.534, *p* < 0.001) and vertebral bodies (γ = 0.514, *p* < 0.001). After adjustment of the within-patient correlation for the total number of lumbar vertebrae, both the conventional BMD and alternative BMD of the vertebrae showed significant positive associations with the ^18^F-fluoride SUVmean of the vertebral corners and bodies, respectively, as shown in Table 3. In addition, there was a significant positive relationship between the ^18^F-fluoride SUVmean of the vertebral corners and bodies (unstandardized β (SE) = 0.711 (0.109), *p* < 0.001).

Table 4 shows the associated factors for the conventional BMD and alternative BMD of the vertebra at each lumbar vertebra level, as assessed using the multilevel mixed-effects linear regression models. After adjusting for the confounding factors, the ^18^F-fluoride SUVmean of the vertebral corners was significantly related with the conventional BMD of the vertebra (unstandardized β (SE) = 0.092 (0.029), *p* = 0.001), but this association was not significant for the ^18^F-fluoride SUVmean of the vertebral bodies (unstandardized β (SE) = 0.069 (0.039), *p* = 0.075). Otherwise, the ^18^F-fluoride SUVmean of both the vertebral corners and vertebral bodies had significant associations with the alternative BMD of the vertebra in the multivariable multilevel mixed-effects linear regression models (unstandardized β (SE) = 0.085 (0.032), *p* = 0.009, and unstandardized β (SE) = 0.114 (0.044), *p* = 0.01, respectively).

Data regarding ^18^F-fluoride SUVmax are presented in Appendix A. Unlike the ^18^F-fluoride SUVmean, the ^18^F-fluoride SUVmax of the vertebrae did not show significant associations with BMD in the multivariable multilevel mixed-effects linear regression models (Appendix A).

## 4. Discussion

The present study investigated the relationship of the regional bone synthetic activities of lumbar vertebral corners and bodies assessed using ^18^F-fluoride PET and with lumbar BMD measured using DXA in patients with AS at each vertebra level. After the adjustment of the within-patient correlation, the ^18^F-fluoride SUVmean of the vertebral corners showed significant associations with both the conventional BMD of the vertebra measured using conventional methods (ROI-1) and the alternative BMD of the vertebra, regarded as trabecular bone area (ROI-2). Otherwise, the ^18^F-fluoride SUVmean of the vertebral bodies was significantly related with the alternative BMD but not with the conventional BMD. In addition, the extents of the ^18^F-fluoride uptake of the vertebral corners and vertebral bodies were significantly associated with each other. ^18^F-fluoride PET/CT in vivo imaging, which uses ^18^F-fluoride as a direct bone tracer, enables the visualization of the molecular change in regional bone metabolism at any skeletal site [25,26]. Therefore, an increased activity of ^18^F-fluoride means increased bone metabolism.

The major result of our study was that the ^18^F-fluoride SUVmean of the vertebral corners was associated with the conventional BMD of the corresponding lumbar vertebrae at each lumbar vertebra level in AS patients. There was a trend towards a positive correlation between the ^18^F-fluoride SUVmean of the vertebral bodies and conventional BMD, but this trend did not reach statistical significance (*p* = 0.071). These findings suggest that the magnitude of the osteoblastic activities in the vertebral corners may be more closely related with the conventional BMD than that of vertebral bodies. Considering that the BMD measured using DXA should provide more information on the status of trabecular bone than that of cortical bone, it is assumed that the ^18^F-fluoride uptake of the vertebral bodies may have more impact on the conventional BMD as compared with that of the vertebral corners. In fact, a previous study found that the ^18^F-fluoride SUV of lumbar vertebral bodies was significantly correlated with the lumbar spine BMD in postmenopausal women [24]. Thus, our finding may be unexpected. Although the exact reasons for our finding are not fully understood, one possible explanation is that the effect of regional bone metabolism in vertebral corners on lumbar spine BMD measured by DXA may be large enough to offset the statistically significant correlation between the ^18^F-fluoride uptake of vertebral bodies and conventional BMD. In any case, it is obvious that conventional BMD measured using DXA in AS patients is considerably affected by the regional bone metabolism at the vertebral corners, and thus, special attention should be paid when interpreting BMD on DXA in these populations. Otherwise, the ^18^F-fluoride uptake of both vertebral corners and vertebral bodies was significantly related with the “alternative BMD” of the vertebra, which is assumed to represent trabecular BMD. This notion suggest that the alternative BMD may more accurately reflect the degree of mineralization of trabecular elements than the BMD measured using conventional methods. However, further studies are needed to validate our methods for evaluating trabecular BMD on DXA by manually drawing ROIs.

There was a significant positive association between baseline ^18^F-fluoride uptake at the vertebral corners and that at the vertebral bodies after adjustment of the within-patient correlation in our study, suggesting a close connection in osteoblastic activities between cortical and trabecular bone in AS. This finding is somewhat unexpected, considering that excessive bone formation at cortical bone and bone loss at trabecular bone can occur in AS, as mentioned above. The ^18^F-fluoride uptake on PET could represent regional bone synthetic activities but not osteoclastic activities, and bone metabolism is determined by a delicate balance between bone formation by osteoblasts and bone resorption by osteoclasts. We assumed that the distinct bone metabolism at cortical and trabecular bone compartments in AS may be largely mediated by osteoclastic activities, which are not detected using ^18^F-fluoride PET. However, additional research investigating the status of bone metabolism at cortical and trabecular bone in AS are necessary to verify our hypothesis.

The main strength of this study is the quantitative measurement of bone metabolism on ^18^F-fluoride PET and BMD on DXA in the separated cortical and trabecular bone compartments of the same lumbar vertebrae in patients with AS. The present study tried to analyze more specified regions of bone formation activities in the vertebral corners and bodies, respectively. It is noteworthy that we newly designed ROIs to analyze the trabecular BMD of the lumbar spine on DXA by manually eliminating the intervertebral area and the edge of each vertebra in order to exclude the impact of sclerotic changes due to new bone formation around cortical bone on BMD in AS. In addition, we were able to measure the ^18^F-fluoride SUVmean of the vertebral corners and bodies separately using spatially identified anatomical sites of the particular interesting regions of the PET images (Figure 1A,B), modifying the previously reported reference [24]. With these ROIs, we could find several significant or unexpected findings between BMDs and SUVmean. Similar to our method, Frost et al. differentiated the regional bone metabolism of hip cortical and trabecular bone using an application of a spatially distinct ROI on ^18^F-fluoride PET images in patients with osteoporosis receiving teriparatide therapy, and the bone synthetic activity of cortical bone was more pronounced than that of trabecular bone during teriparatide treatment [13]. Taken together, the measurement of regional bone metabolism ^18^F-fluoride PET can provide a better insight into the pathophysiology of metabolic bone diseases including AS.

The present study has several limitations. First, we used CT, which was not performed with PET simultaneously. The MR-based attenuation map for PET reconstruction posed challenges regarding underestimating the SUV of the bone, especially the cortical area, because of its high attenuation [27,28]. Therefore, we decided that CT was more appropriate for the attenuation correction for this study to quantify the activity of ^18^F-fluoride in bone regions of interest. Second, the lateral DXA would be more appropriate compared with the PET image of sagittal plane; however, the current study only used the existing data from the previously performed prospective study. Therefore, we tried to eliminate the cortical area in the AP view of DXA as much as possible, as depicted in Figure 1D. Third, we could not use the trabecular bone score (TBS), a recent developed method for the evaluation of the microarchitecture of the lumbar spine [29]. Though it is known that the TBS can be extracted from the existing DXA images, the DXA data included in the present study were out of date for analyzing the TBS. Fourth, due to the study’s exploratory nature, we analyzed a small number of patients and lumbar vertebrae. Thus, further larger investigations are necessary to confirm our findings.

## 5. Conclusions

In conclusion, the bone synthetic activities of vertebral corners were more closely related with BMD on DXA at each vertebra level in patients with AS than with those of the vertebral bodies, suggesting that the effects of regional bone metabolism at the vertebral corners and bodies on BMD may differ in AS. Otherwise, the ^18^F-fluoride uptake of the vertebral corners and vertebral bodies were significantly associated with an alternative BMD, which is assumed to represent the status of the trabecular bone. Our data also revealed that there was a significant correlation between the magnitude of the osteoblastic activities between vertebral bodies and vertebral corners of lumbar spine in patients with AS. We believe that this study could provide novel insight into bone metabolism in AS, which affects cortical and trabecular bone differently, and highlight a potential usefulness of ^18^F-fluoride PET in investigating the underlying mechanism of structural changes in AS.

## Figures and Tables

**Figure 1 jcm-09-02656-f001:**
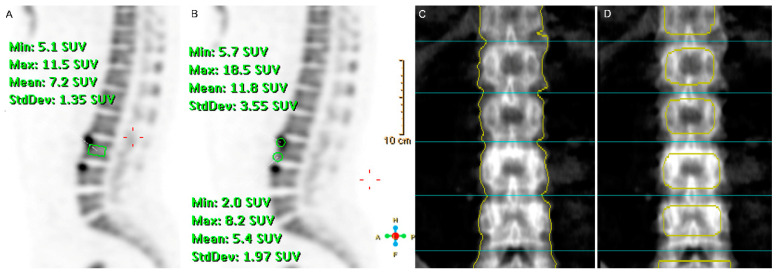
A representative example of a ^18^F-fluoride positron emission tomography (PET) image and dual energy X-ray absorptiometry (DXA) for a 42-year old male patient with ankylosing spondylitis. The region of interest (ROI) used to obtain the mean standardized uptake value of the L1 to L4 in the mid-sagittal plane of the ^18^F-fluoride PET image. The rectangular region of interest (ROI) for the vertebral body (**A**); average volxel size ± SD (981.84 ± 0.14 mm^3^) and 2 circle-shaped ROIs with 1 cm diameter for calculating the arithmetic mean standardized uptake values (SUV)mean of the upper and lower vertebral corners (**B**), average area ± SD; average volxel size ± SD (324.14 ± 0.00 mm^3^). For generating the different ROIs for calculating the bone mineral density (BMD) of L1–L4, the ROI-1 (**C**) average BMD ± SD (1.33 ± 0.001 g/cm^2^) of each vertebra was acquired according to a routinely used method, and the ROI-2 (**D**) average BMD ± SD (1.23 ± 0.001 g/cm^2^) was manually drawn to exclude the cortical bone area on the projection image. BMD measured using ROI-1 and ROI-2 was termed as “conventional BMD” and “alterative BMD”, respectively.

**Figure 2 jcm-09-02656-f002:**
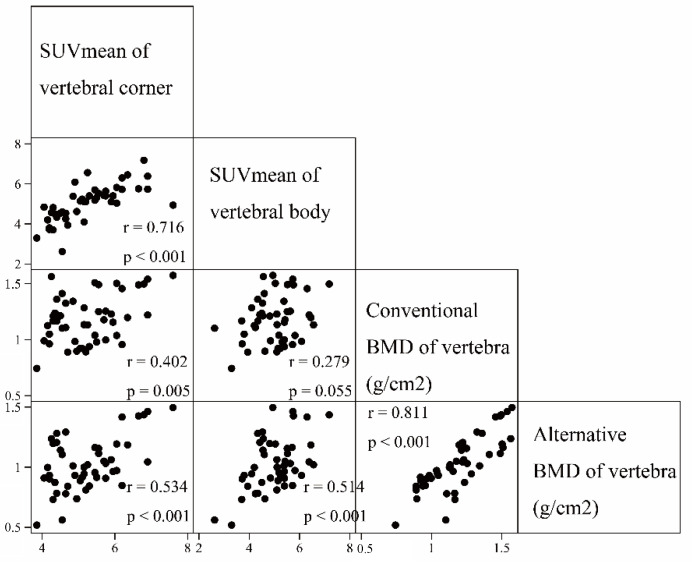
Correlation plots between baseline ^18^F-fluoride mean standardized uptake values of the vertebral bodies and corners, and conventional and alternative bone mineral density at each vertebra level.

**Table 1 jcm-09-02656-t001:** Demographic and clinical data in patients with ankylosing spondylitis.

Characteristics	Patients with AS (*n* = 12)
Males, *n* (%)	12 (100)
Age (years), median (IQR)	42 (32–42.8)
Disease duration (years), median (IQR)	7.5 (2.4–10.1)
HLA-B27 positive, *n* (%)	12 (100)
mSASSS, median (IQR)	9 (5.3–28)
BMI (kg/m^2^), median (IQR),	26.3 (22.9–27.6)
ESR (mm/hr), median (IQR)	13 (5.8–20)
CRP (mg/dL), median (IQR)	0.33 (0.06–0.6)
BASDAI, median (IQR)	3.5 (1.2–6)
BASFI, median (IQR)	2.4 (0.7–3.8)
BASMI, median (IQR)	2.8 (1.3–4.4)
NSAID use, *n* (%)	12 (100)
SSZ use, *n* (%)	12 (100)
TNF-α inhibitor use, *n* (%)	0 (0)

AS: ankylosing spondylitis, mSASSS: modified Stoke Ankylosing Spondylitis Spine Score, BMI: body mass index, ESR: erythrocyte sedimentation rate, CRP: C-reactive protein; BASDAI: Bath Ankylosing Spondylitis Disease Activity Index, BASFI: Bath Ankylosing Spondylitis Functional Index; BASMI: Bath Ankylosing Spondylitis Metrology Index, NSAIDSs: non-steroidal anti-inflammatory drugs, SSZ: sulfasalazine, MTX: methotrexate, TNF-α: tumor necrosis factor-α.

**Table 2 jcm-09-02656-t002:** The magnitude of ^18^F-fluoride uptake on positron emission tomography and bone mineral density on dual energy X ray absorptiometry of lumbar spine in patients with ankylosing spondylitis at each vertebra level.

	Lumbar Vertebrae (*n* = 48)
^18^F-Fluoride SUVmean of vertebral corner ^a^, mean ± SD	5.24 ± 0.9
^18^F-fluoride SUVmean of vertebral body, mean ± SD	5.03 ± 0.9
Conventional BMD of vertebra (g/cm^2^), mean ± SD	1.19 ± 0.21
Alternative BMD of vertebra (g/cm^2^), mean ± SD	1.03 ± 0.22

^a^ This indicates the arithmetic mean SUVmean of the upper and lower vertebral corners. SUVmean: mean standardized uptake values, BMD: bone mineral density.

**Table 3 jcm-09-02656-t003:** Relationship between bone mineral density and ^18^F-fluoride mean standardized uptake value of each vertebra analyzed by univariable multilevel mixed-effects linear regression, adjusting for within-patient correlations for the total number of lumbar vertebrae.

Dependent Variable	Independent Variable	Univariable Model	
		Unstandardized β (SE)	*p* Value
Conventional BMD of vertebra (g/cm^2^)	^18^F-fluoride SUVmean of vertebral corner ^a^	0.112 (0.028)	<0.001
Conventional BMD of vertebra (g/cm^2^)	^18^F-fluoride SUVmean of vertebral body	0.103 (0.042)	0.014
Alternative BMD of vertebra (g/cm^2^)	^18^F-fluoride SUVmean of vertebral corner ^a^	0.123 (0.031)	<0.001
Alternative BMD of vertebra (g/cm^2^)	^18^F-fluoride SUVmean of vertebral body	0.158 (0.041)	<0.001
^18^F-fluoride SUVmean of vertebral corner ^a^	^18^F-fluoride SUVmean of vertebral body	0.711 (0.102)	<0.001

^a^ This indicates the arithmetic mean SUVmean of the upper and lower vertebral corners. BMD: bone mineral density, SUVmean: mean standardized uptake values.

**Table 4 jcm-09-02656-t004:** Associated factors for bone mineral density at each lumbar vertebra level assessed by multilevel mixed-effects linear regression, adjusting for within-patient correlations for the total number of vertebrae.

Dependent Variable	Independent Variable	Univariable Model		Multivariable Model ^b^	
		Unstandardized β (SE)	*p* Value	Unstandardized β (SE)	*p* Value
Conventional BMD (g/cm^2^)	^18^F-fluoride SUVmean of vertebral corner ^a^	0.112 (0.028)	<0.001	0.092 (0.029)	0.001
	^18^F-fluoride SUVmean of vertebral body	0.103(0.042)	0.014	0.069 (0.039)	0.075
	Disease duration (months)	0.002 (0.001)	0.025	0.002 (0.001)	0.038
	CRP (mg/dL)	−0.153 (0.106)	0.146	−0.164 (0.091)	0.071
	Age (years)	−0.001 (0.007)	0.854		
	BMI (kg/m^2^)	0.013 (0.013)	0.338		
	BASDAI	−0.017 (0.023)	0.467		
Alternative BMD (g/cm^2^)	^18^F-fluoride SUVmean of vertebral corner ^a^	0.123 (0.031)	<0.001	0.085 (0.032)	0.009
	^18^F-fluoride SUVmean of vertebral body	0.158 (0.041)	<0.001	0.114 (0.044)	0.01
	BMI (kg/m^2^)	0.017 (0.013)	0.187	0.005 (0.013)	0.702
	Disease duration (months)	0.001 (0.001)	0.585		
	CRP (mg/dL)	−0.067 (0.104)	0.5558		
	Age (years)	−0.006 (0.007)	0.35		
	BASDAI	−0.004 (0.024)	0.877		

^a^ This indicates the arithmetic mean SUVmean of the upper and lower vertebral corners. ^b^ Estimated using multivariable multilevel mixed-effects linear regression models including the ^18^F-fluoride SUVmean of the vertebral body and vertebral corners, disease duration and CRP for conventional BMD and including the ^18^F-fluoride SUVmean of the vertebral body and vertebral corners and BMI for alternative BMD, respectively. SE: standard error, BMD: bone mineral density, SUVmean: mean standardized uptake values, CRP: C-reactive protein, BMI: body mass index, BASDAI: Bath Ankylosing Spondylitis Disease Activity Index.

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
