# Peer review of "Association of Regional Bone Synthetic Activities of Vertebral Corners and Vertebral Bodies Quantified Using 18F-Fluoride Positron Emission Tomography with Bone Mineral Density on Dual Energy X-ray Absorptiometry in Patients with Ankylosing Spondylitis"

_jcm, 2020, doi:10.3390/jcm9082656_

Round 1

Reviewer 1 Report

Interesting paper in the field of bone quantification. 

My main remark is the relevance of comparing bone quantification by F-18-fluoride PET and bone density on DXA in regions of interest (ROIs) that are not comparable. Indeed the regions of interest on the F-18-fluoride PET are three dimensional regions of interest and the regions of interest on DXA two dimensional regions of interest with overlap of several structures that are not taken into account in the PET ROIs. My question to the authors is if they could measure bone density in the same region of interest as in the PET on the CT-scan, which is more precise in localizing corresponding bone metabolism, and would help further document regional differences in bone metabolism. If not could they explain the rational and clinical interest?

Author Response

: Thank you for your comments. If we use CT for the quantification of BMD, quantitative CT(qCT) is required. Only using CT, we only acquired the HU which implied the relative density of bone lesion and HU of CT is affected by the attenuation and the use of contrast. BMD value as expressed as g/cm2 of DXA or mg/cm3 of qCT are the absolute quantitative values and we thought that existing BMD value from DXA would be appropriate for the purpose of our study design. Though DXA is a projection image and there is inherent limitation for to separate the trabecular and cortical bone areas, we tried to eliminate the cortical bone area manually drawing new region of interest as depicted in figure 1.

Reviewer 2 Report

  In this manuscript, the authors investigated whether the osteoblastic bone synthesis activities of the lumbar vertebrae quantified by 18F-fluoride PET is associated with BMD of the corresponding vertebrae quantified by DXA. 

  The authors make a commendable effort to statistically demonstrate the association of PET SUV values and DXA BMD values, but this reviewer has significant concerns regarding the design and methodology of this study. Since the DXA employed in this study examined L1-L4, the PET SUV values from the corresponding vertebra were compared. 18F-fluoride PET SUV from the vertebral body and the averaged SUV from the superior and inferior anterior corners on a mid-sagittal image were measured and analyzed. 

  Although I am not a statistician, I do not think that values from different vertebra of the same patient can be considered to be independent variables, so I highly recommend that the authors discuss this issue with a statistician and describe how they approached this problem in more detail. I am also skeptical about running linear regression analyses with small number of samples, so I ask the authors to also consider this issue.

  More importantly, the active lesions with increased 18F-fluoride uptake are not uniform throughout the lumbar vertebrae of patients with AS, as is illustrated in Figure 1. All intervertebral segments may be afflicted, it may be only one segment, or somewhere between. The authors may feel that a uniform examination of all vertebrae may average out the differences, but I am uncomfortable with this design. The authors claim to account for the clustering of variables from the L1 to L4 vertebrae of each patient by utilizing a generalized estimating equation model, but that is a type of linear regression tool that cannot account for the discrepancies between active and non-active segments. It may be more prudent to pick the vertebra with the greatest 18F-fluoride uptake and perform the analysis with BMD of that vertebra.

  On a technical note, for a true BMD of the vertebral body, a lateral DXA of the lumbar spine with ROI placed within the cortical outlines of the vertebral body is preferable. An AP DXA with ROI as depicted in Fig. 1D will still affected by the influences from the cortices of the pedicles and spinous processes.

  18F-fluoride uptake has been shown to reflect osteoblastic bone synthesis activity, but it has also been shown to be strongly influenced by inflammation. The authors only make a glancing reference to this fact, but this needs to be explained in further detail. 

  More fundamentally, I am having difficulty understanding why the authors set out to examine whether there was an association with the osteoblastic bone synthesis activities of AS lesion sites and the bone mineral density of the affected lumbar vertebrae, especially with the vertebral body. The increased 18F-fluoride uptake reflects the inflammatory and osteoblastic bone synthesis activity of the vertebral body edges that leads to syndesmophyte development, as stated in the authors' past publications. As such, it is thought to increase and decrease according to the severity of the AS pathological process at the time of the scan. The BMD of vertebrae reflects the total bone metabolism process of that patient up until the DXA examination, and can be regarded as a more accumulative value. Therefore, this reviewer does not believe that it is fundamentally correct to look for an association between these indices. It is akin to looking for an association between the speed recorded of a moving car at one intersection and the time it took to travel across the continental United States. Yes, a faster driver may have higher values in both indices, but how would you account for stops taken along the route? 

Author Response

Although I am not a statistician, I do not think that values from different vertebra of the same patient can be considered to be independent variables, so I highly recommend that the authors discuss this issue with a statistician and describe how they approached this problem in more detail. I am also skeptical about running linear regression analyses with small number of samples, so I ask the authors to also consider this issue.

: Thank you for your kind comment. Several previous studies also analyzed the verterbra level data in patients with AS using generalized estimating equation model. We cited representative papers regarding vertebra level data in AS (Reference 25 and 26). We analyzed 48 lumbar vertebrae and think that this number is enough to conduct linear regression analyses. In general, if the number of samples is 30 or more, this variable had a normal distribution and it is possible to perform a linear regression analysis using this variable as a dependent as well as independent variables.

More importantly, the active lesions with increased 18F-fluoride uptake are not uniform throughout the lumbar vertebrae of patients with AS, as is illustrated in Figure 1. All intervertebral segments may be afflicted, it may be only one segment, or somewhere between.

: We appreciate you for pointing out an important consideration. Because the each vertebra shows the different stage of disease progression in AS patients and we thought that the Figure 1 represents such characteristics of AS patients well. The linear correlation between the BMD values and F-18 NaF activity also reflect the feature of each vertebral pathologic process of the AS patients. In addition, we refereed to the CT images to draw ROI, meticulously to exclude the non-interested regions, such as intervertebral or paravertebral activities.

The authors may feel that a uniform examination of all vertebrae may average out the differences, but I am uncomfortable with this design. The authors claim to account for the clustering of variables from the L1 to L4 vertebrae of each patient by utilizing a generalized estimating equation model, but that is a type of linear regression tool that cannot account for the discrepancies between active and non-active segments. It may be more prudent to pick the vertebra with the greatest 18F-fluoride uptake and perform the analysis with BMD of that vertebra.

: In AS, each vertebra has its own pathological features and processes, which may not be reflected by analysis of the association between the vertebra with the greatest 18F-fluoride uptake and BMD of corresponding vertebra (your suggestion). As mentioned above, GEE has been considered as appropriate statistical method when analyzing imaging data of AS like ours and many previous studies used GEE. Generalized estimating equation (GEE) is used to estimate the parameters of a generalized linear model with a possible unknown correlation between outcomes and are commonly used in large epidemiological studies, especially multi-site cohort studies, clustered/ hierarchical data and imaging data of AS which has within patient correlation.

In addition, we also analyzed our data using SUVmax of vertebra instead of SUVmean and presented the results as supplementary tables and figure in the revised manuscript. SUVmax is appropriate rational in oncology and SUVmean is used in musculoskeletal diseases.

On a technical note, for a true BMD of the vertebral body, a lateral DXA of the lumbar spine with ROI placed within the cortical outlines of the vertebral body is preferable. An AP DXA with ROI as depicted in Fig. 1D will still affected by the influences from the cortices of the pedicles and spinous processes.

: Thank you for your comments. We agree with your comment that the lateral DXA would be more appropriate to be compared with the F-18 NaF activity at mid-sagittal plane of PET. However, this study was a post hoc analysis of previously performed prospective study and DXA was performed for the evaluation of BMD of patients not for the purpose of research. The lateral view of L-spine with DXA is not standard method for the evaluation of BMD status for the patients according to the ISCD guideline. In addition, though we could acquire the lateral DXA, lateral DXA also include the lateral cortical bone area because DXA is a projection image. Therefore, we tried to use the predisposing imaging data of AP view of DXA and to eliminate the cortical bone area as drawing ROIs as depicted in Fig. 1D. We described about this as a limitation in the discussion section.

 18F-fluoride uptake has been shown to reflect osteoblastic bone synthesis activity, but it has also been shown to be strongly influenced by inflammation. The authors only make a glancing reference to this fact, but this needs to be explained in further detail.

More fundamentally, I am having difficulty understanding why the authors set out to examine whether there was an association with the osteoblastic bone synthesis activities of AS lesion sites and the bone mineral density of the affected lumbar vertebrae, especially with the vertebral body. The increased 18F-fluoride uptake reflects the inflammatory and osteoblastic bone synthesis activity of the vertebral body edges that leads to syndesmophyte development, as stated in the authors' past publications. As such, it is thought to increase and decrease according to the severity of the AS pathological process at the time of the scan. The BMD of vertebrae reflects the total bone metabolism process of that patient up until the DXA examination, and can be regarded as a more accumulative value. Therefore, this reviewer does not believe that it is fundamentally correct to look for an association between these indices. It is akin to looking for an association between the speed recorded of a moving car at one intersection and the time it took to travel across the continental United States. Yes, a faster driver may have higher values in both indices, but how would you account for stops taken along the route?

: Although our previous studies concluded that 18F-fluoride of vertebral corner could reflect both inflammatory and osteoblastic bone synthesis activity in AS patients, 18F-fluoride PET has been generally considered as a tool representing osteoblastic activity. In addition, the magnitude of statistical significance was greater for bone synthetic activity compared with inflammation in our previous study. As mentioned in introduction section, inflammation and bone formation are known to be linked with each other in AS. As you commented, 18F-fluoride PET may not reflect osteoblastic activity only, but we believe that this imaging modality could provide a novel insight into bone metabolism in AS.

Reviewer 3 Report

Comments to the Author

The manuscript examined the association between the bone synthetic activities of vertebral bodies or vertebral corners quantified using 18F-fluoride positron emission tomography (PET) and bone mineral density (BMD) at the lumbar vertebrae in ankylosing spondylitis (AS). The results give us useful knowledge. However, revision is needed for acceptance.

My comments are following:

  1. As to the clinical usefulness, I'm not very familiar with ankylosing spondylitis. Is the PET image a routine inspection for AS? If not, you should describe why you used the PET in more detail.

Author Response

: Thank you for your comment. PET images is not routine examination for AS patients in real clinical practice. 18F-fluoride PET was performed for research purposes in our study. As mentioned in introduction section, because the regional bone metabolism at different sites can be measured by 18F-fluoride PET, we hypothesized that 18F-fluoride PET may be a useful tool for the assessment of bone metabolism of AS. Detailed information regarding PET was described in the introduction and discussion sections.

Reviewer 4 Report

In the results SUV max should be given !

The number of pts should be enlarged  

Author Response

In the results SUV max should be given !

: We appreciate your kind comment. We added the results of SUVmax in results section and supplementary tables and figure. Unlike 18F-fluoride SUVmean, 18F-fluoride SUVmax of vertebrae did not show significant associations with BMD in multivariable GEE models (Supplementary Table 3).

The number of pts should be enlarged

: We agree with the authors’ opinion that we need data from different cohorts to develop a prediction model and then need to validate it for the prediction of an outcome in clinical practice such as response in patients with AS. As we are constantly acquiring F-18 NaF PET/CT for AS patients, we could report more reliable results from the larger data following the authors’ consideration in the future.

Round 2

Reviewer 1 Report

The authors have provided an adequate answer to my question.

Author Response

We appreciate your kind comment.

Reviewer 2 Report

  The authors have provided a response to the points that I have raised, but I still have strong reservations as to the design and statistical integrity of this study. 

  First and foremost, the application of a generalized estimating equation (GEE) to analyze the relationship between the 18F-fluoride uptake and BMD of each vertebra remains highly questionable.

  The authors have claimed that GEE is considered to be an appropriate statistical method when analyzing imaging data of AS and has been performed in many previous studies. The reference that the authors point to performs a longitudinal study on MRI findings and new bone formation, with GEE being used to perform longitudinal comparisons of binomial outcomes. I can understand the use of the GEE method to analyze longitudinal data, especially when constricted to binomial variables. However, I remain unpersuaded that GEE is the proper method to analyze this situation with a one-time analysis of continue variables. 

  Examining the assumptions that are required before performing a GEE analysis, the foremost condition is independence of the the covariates. The authors reply that they analyzed 48 lumbar vertebrae and think that it is enough to conduct linear regression analyses. I would agree if the authors had 48 independent vertebrae to analyze. Unfortunately, the data is comprised from 4 lumbar vertebrae from 12 patients. This is an example of correlated observations, and needs to be analyzed by within-cluster comparisons such as hierarchical linear models; analysis with a linear regression leads to an underestimation of p values. 

  I understand that the authors' previously reported that 18F-fluoride of vertebral corner reflects osteoblastic bone synthesis activity more than inflammatory activity in AS patients. However, that does not answer my previous comment asking the authors to explain why they think that 18F-fluoride activity at one time point would be associated with BMD, which reflects the total bone metabolism process of that patient's life up until the DXA examination. I assume that the 18F-fluoride activity is not constant during an AS patients' long bout with this illness. Do you think the vertebra BMD actually increases or decreases with 18F-fluoride activity? The authors need to provide references of the long-term time course of 18F-fluoride activity if they wish to demonstrate that a one-time 18F-fluoride activity reading would affect BMD. The authors also need to discuss the different levels of BMD seen in each patient, comparing vertebrae with low and high levels of 18F-fluoride activity.

Author Response

 "The authors have claimed that GEE is considered to be an appropriate statistical method when analyzing imaging data of AS and has been performed in many previous studies. The reference that the authors point to performs a longitudinal study on MRI findings and new bone formation, with GEE being used to perform longitudinal comparisons of binomial outcomes. I can understand the use of the GEE method to analyze longitudinal data, especially when constricted to binomial variables. However, I remain unpersuaded that GEE is the proper method to analyze this situation with a one-time analysis of continue variables.

 Examining the assumptions that are required before performing a GEE analysis, the foremost condition is independence of the the covariates. The authors reply that they analyzed 48 lumbar vertebrae and think that it is enough to conduct linear regression analyses. I would agree if the authors had 48 independent vertebrae to analyze. Unfortunately, the data is comprised from 4 lumbar vertebrae from 12 patients. This is an example of correlated observations, and needs to be analyzed by within-cluster comparisons such as hierarchical linear models; analysis with a linear regression leads to an underestimation of p values."

: We appreciated your kind comment. As your recommendation, we re-analyzed our data using multilevel mixed-effects linear regression models in order to adjust within-cluster correlations. We used ‘xtmixed’ command in STATA 11.1. Accordingly, we amended methods and results section, Table 3, 4 and supplementary Table 2 and 3 (highlighted in yellow). Multilevel models (also known as hierarchical linear models, linear mixed-effect model, mixed models, nested data models, random coefficient, random-effects models, random parameter models, or split-plot designs) are statistical models of parameters that vary at more than one level such as our data.

"I understand that the authors' previously reported that 18F-fluoride of vertebral corner reflects osteoblastic bone synthesis activity more than inflammatory activity in AS patients. However, that does not answer my previous comment asking the authors to explain why they think that 18F-fluoride activity at one time point would be associated with BMD, which reflects the total bone metabolism process of that patient's life up until the DXA examination. I assume that the 18F-fluoride activity is not constant during an AS patients' long bout with this illness. Do you think the vertebra BMD actually increases or decreases with 18F-fluoride activity? The authors need to provide references of the long-term time course of 18F-fluoride activity if they wish to demonstrate that a one-time 18F-fluoride activity reading would affect BMD."

: It is known that BMD of AS patients can vary over time. BMD of AS patients can decrease with age, however, certain treatment such as TNF blockers can increase BMD (ref. Bone Mineral Density and Fracture Risk in Ankylosing Spondylitis: A Meta-Analysis. Calcif Tissue Int. 2017;101:182-192, Effect of TNF-alpha inhibitor treatment on bone mineral density in patients with ankylosing spondylitis: a systematic review and meta-analysis. Semin Arthritis Rheum 2014;44:155-61). The imbalance of the bone metabolism also might deteriorate the final BMD of AS patients and 18F-fluoride PET is a noninvasive imaging biomarker which is able to provide quantified value of abnormal osteoblastic changes.

As your comment, vertebral BMD could be the result of the accumulation of life-long bone metabolism. However, the enrolled patients in the present study were relatively too young to discuss about the life-long outcome of BMD. We tried to find out the correlation between quantified values of PET and BMD values and the result of our study showed that AS patients had regional different metabolism which leads the regional different BMD values. We thought that BMD values in the current study did not suggest the final result of the patients as the BMD values showed diverse in each vertebra of each patient. We thought that our results meant that the process of bone metabolism not the final outcome of the bone change in AS patients.

In addition, we also bring up a problem in acquisition of image of DXA. Because the pathophysiology of AS is one of abnormal bone metabolism characterized by pathological new bone formation in the cortical zones of the vertebrae with loss of trabecular bone in the center of the vertebral bodies.

To our literature review, there is no study investigating long-term time course of 18F-fluoride activity of axial skeleton in patients with AS. Thus, it is not clear whether the 18F-fluoride activity of axial skeleton in patients with AS is constant of not. To demonstrate your comment, we think that further longitudinal studies are obviously needed.

"The authors also need to discuss the different levels of BMD seen in each patient, comparing vertebrae with low and high levels of 18F-fluoride activity."

: Unlike general population, the structural changes and bone metabolism that occurs in each vertebra are quite different in a same patient with AS (for example, L1 vertebra has new bone formation such as syndesmophyte but L2 vertebra has no structural bone changes and compression fracture in a same patient). Accordingly, a same patient can have variable or different BMD of each vertebra. Thus, the primary interest of our study is to analyze the association between osteoblastic activity on PET and BMD on DEXA at “each vertebra level” in AS patients. As mentioned above, we think that our multilevel mixed-effects linear regression models can analyze the association BMD and 18F-fluoride activity.  

Reviewer 4 Report

Thank you - Revision is ok with me

Author Response

We appreciate your comment.